# Comparative Proteomics Study on the Postharvest Senescence of *Volvariella volvacea*

**DOI:** 10.3390/jof8080819

**Published:** 2022-08-04

**Authors:** Lei Zha, Mingjie Chen, Qian Guo, Zongjun Tong, Zhengpeng Li, Changxia Yu, Huanling Yang, Yan Zhao

**Affiliations:** Institute of Edible Fungi, Shanghai Academy of Agricultural Sciences, Shanghai 201403, China

**Keywords:** *V. volvacea*, postharvest, senescence, comparative proteomics

## Abstract

*Volvariella volvacea* is difficult to store after harvest, which restricts the production and circulation of *V. volvacea* fruiting bodies. Low-temperature storage is the traditional storage method used for most edible fungi. However, *V. volvacea* undergoes autolysis at low temperatures. When fruiting bodies are stored at 15 °C (suitable temperature), *V. volvacea* achieves the best fresh-keeping effect. However, the molecular mechanism underlying the postharvest senescence of *V. volvacea* remains unclear. Based on this information, we stored *V. volvacea* fruiting bodies at 15 °C after harvest and then analyzed the texture and phenotype combined with the results of previous physiological research. Four time points (0, 24, 60, and 96 h) were selected for the comparative proteomics study of *V. volvacea* during storage at 15 °C. A variety of proteins showed differential expressions in postharvest *V. volvacea* at 15 °C. Further comparison of the gene ontology (GO) enrichment analysis and KEGG pathways performed at different sampling points revealed proteins that were significantly enriched at several time points. At the same time, we also analyzed differentially expressed proteins (DEPs) related to the RNA transport, fatty acid biosynthesis and metabolism, and amino acid biosynthesis and metabolism pathways, and discussed the molecular functions of the PAB1, RPG1, ACC1, ADH3, ADH2, ALD5, and SDH2 proteins in postharvest *V. volvacea* senescence. Our results showed that many biological processes of the postharvest senescence of *V. volvacea* changed. Most importantly, we found that most RNA transport-related proteins were down-regulated, which may lead to a decrease in related gene regulation. Our results also showed that the expression of other important proteins, such as the fatty acid metabolism related proteins increased; and changes in fatty acid composition affected the cell membrane, which may accelerate the ripening and perception of *V. volvacea* fruiting bodies. Therefore, our research provides a reference for further studies on the aging mechanism of *V. volvacea*.

## 1. Introduction

Senescence is the final stage of fruit development and is accompanied by drastic changes in fruit flavor, peel color, texture, pathogen sensitivity, and biochemical composition [1,2,3]. The postharvest senescence of the fruiting bodies of edible fungi seriously affects their commercial value. *V. volvacea*, also called Chinese mushroom, is mainly cultivated in tropical and subtropical regions, and is a typical, high-temperature edible fungus suitable for growing in a high-temperature environment [4]. Delicious *V. volvacea* fruiting bodies are mainly sold fresh [5]. However, the fruiting bodies of *V. volvacea* actively undergo physiological metabolism and easily age under normal storage temperatures. In addition, the storage of *V. volvacea* below 10 °C easily leads to the autolysis of the fruiting bodies [6,7]. Alternatively, the storage of fresh *V. volvacea* at 25 °C increases its respiratory rate and metabolic activity, and the quality of fresh *V. volvacea* fruiting bodies decreases rapidly within 48 h of storage, thus losing commercial value. Meanwhile, researchers found that the best storage of *V. volvacea* was achieved when the temperature was approximately 15 °C [8].

Although postharvest storage at 15 °C improves the storage quality of *V. volvacea*, a suitable temperature is insufficient if we want to achieve the long-distance transportation, market circulation, and maximum commercial value of *V. volvacea*. However, we do not yet have a way to address the problems of *V. volvacea* storage. In addition, irradiation [9], air conditioning [10], the application of preservatives [11], biological preservation [12,13], and other methods are often used to increase the storage time of *V. volvacea*. Although the storage period of *V. volvacea* can be extended to a certain extent, its proper postharvest storage conditions cannot be fundamentally determined due to an insufficient understanding of the internal mechanism of the postharvest aging of *V. volvacea*. The postharvest commercialization of fruits and vegetables involves the complex regulation of the postharvest ripening and aging process of fruits and vegetables, such as the biosynthesis of proteins, pigments, and volatile compounds, the accumulation of sugars and organic acids, and other physiological and biochemical changes. These changes are affected not only by commercial treatments, such as postharvest preservation, storage, and transportation conditions, but also by the expression of genes and proteins in the fruits and vegetables themselves.

Proteomics is a powerful tool to study changes in the protein composition and abundance during fruit ripening and the effect of post-harvest treatment on the proteome of the fruit [14]. Proteomics also provides a new method for studying edible fungi, and proteomics research on edible fungi has achieved great progress. Edible fungi that have been studied by proteomics include *Lentinula edodes* [15], *Boletus edulis* [16], *Flammulina velutipes* [17,18], *Pleurotus tuberregium* [19], *Termitomyces heimii* [20], *Sparassis crispa*, and *Hericium erinaceus* [21]. Advances in proteomics technology have enabled researchers to quantitatively measure proteomic changes in complex samples. In particular, the isobaric tags for relative and absolute quantitation (iTRAQ)-based quantitative proteomics have become a powerful tool of choice, because they cover a variety of proteins due to limitations in molecular weight, isoelectric point, or hydrophobicity. At the same time, iTRAQ technology has not been used in the evaluation of the postharvest senescence of *V. volvacea* at 15 °C.

Therefore, fruiting bodies of *V. volvacea* were selected to be stored at 15 °C. Based on our previous study [22], in which the changes in the morphological and physiological indexes at 0, 12, 24, 36, 48, 60, 72, and 96 h were measured, four time points, namely 0, 24, 60, and 96 h, were selected for this comparative proteomics study utilizing iTRAQ technology. From the perspective of proteomics, the dynamic changes in the proteome during the postharvest storage of *V. volvacea* at 15 °C were revealed, which provided a theoretical basis for the development of enzyme inhibitors to inhibit the aging of *V. volvacea*, thereby prolonging the fresh-keeping period of this mushroom. At the same time, our results will provide a new perspective for studying the molecular mechanism of *V. volvacea* postharvest storage at 15 °C.

## 2. Materials and Methods

### 2.1. V. volvacea Samples

Samples of *V. volvacea* V23 were collected from the Shanghai Fanshun Edible Fungi Professional Cooperative and immediately sent to the lab of the Shanghai Academy of Agricultural Sciences for treatment. We selected full, smooth-surfaced, and disease-free fruiting bodies, and the size of the complete fruiting bodies was approximately the same across samples. The samples were randomly divided into four groups (3 fruit bodies in each group), with 3 replicates in each group. The samples were stored at 15 °C for 0, 24, 60, and 96 h (labeled S0, S24, S60, and S96, respectively).

### 2.2. Physiological, Phenotypic, and Texture Experiments

This study included a preliminary examination of some physiological indexes of *V. volvacea* stored at 15 °C after harvest. The weight loss rate (WLR), relative electric conductivity (REC), and malondialdehyde (MDA) content were determined using the research methods and data reported by Zha et al. [22]. The texture was measured on the top of the *V. volvacea* fruiting bodies with a texture analyzer [23].

### 2.3. Protein Preparation, Protein Digestion, and iTRAQ Labeling

The protein extraction methods were followed as described in a previous study [24]. Protein digestion and iTRAQ labeling were performed in accordance with the procedure described by Wisniewski et al. and Zha et al. [5,25]. The protein extraction, protein digestion, and iTRAQ labeling experimental steps are described in Zha et al. [5].

### 2.4. Reverse-Phase Liquid Chromatography (RPLC) and MS Analysis

For the reverse-phase liquid chromatography (RPLC) analysis, RP separation was performed on a 1100 HPLC system (Agilent Technologies, Santa Clara, CA, USA) using an Agilent Zorbax Extended RP column (5 μm, 150 mm × 2.1 mm). All analyses were performed using a Q Exactive mass spectrometer equipped with a Nanospray Flex Ion Source (Thermo Fisher Scientific, Waltham, MA, USA). The methods are described in Zha et al. [5].

### 2.5. Protein Identification and Quantification

All the TripleTOF 5600 MS/MS raw data were thoroughly searched against the sample protein database using ProteinPilot software (v.5.0) (AB Sciex, Framingham, MA, USA). The search database was the *V. volvacea* database (https://mycocosm.jgi.doe.gov/Volvo1/Volvo1.home.html) (accessed on 7 October 2019). This database was used for protein retrieval. A global false discovery rate (FDR) based on fit ≤ 1% and peptides ≥ 2 was the criterion for a reliable protein evaluation. Based on the selected reliable protein, Student’s t-test was used to identify statistically significant changes in proteins (*p* < 0.05). Fold changes (FC) were calculated relatively for two comparison groups (S96, S60, S24, compared to S0). For each comparison, the protein with a FC of >1.5 (or <2/3) was considered to be a differentially abundant protein.

### 2.6. Bioinformatics and Statistical Analyses

GO (Gene Ontology) and KEGG (Kyoto Encyclopedia of Genes and Genomes) analyses were used to enrich the DEPs with OmicsBean software (http://www.omicsbean.cn) (accessed on 18 December 2019) [26].

## 3. Results

### 3.1. Physiological, Phenotypic, and Texture Analyses

The results [22] showed that the WLR (Appendix A), REC (Appendix A), and MDA (Appendix A) increased with storage time. The hardness is the maximum force that a sample material resists pressing into its surface, which reflects the force on *V. volvacea* without deformation. It is the most direct indicator of taste. *V. volvacea* samples with high hardness and low cohesiveness can show fracturability. The texture results showed that hardness was significant during 60–96 h (Figure 1A) and fracturability was significant at 96 h (Figure 1B). The structure of the mushroom changed significantly at 60–96 h relative to that at the beginning of the storage period. The appearance of the fruiting bodies of *V. volvacea* did not change noticeably during the 24 h storage period (Figure 1C). With increasing storage time, the surface of the fruiting bodies browned. Rot became serious and browning was aggravated, among other changes. This may be caused by the loss of water in the fruiting body, the destruction of cell membranes, and the decline of self-regulation during the postharvest storage of *V. volvacea*.

### 3.2. Identification of DEPs

An iTRAQ-based proteomic quantitative analysis was performed on samples collected at 0, 24, 60, and 96 h of storage to understand the proteomic changes in the *V. volvacea* fruiting bodies during storage at 15 °C. Notably, 2063 reliable proteins were identified. The proteomes of S24–S0, S60–S0, and S96–S0 were compared. Protein abundances with a fold change (FC) of ≥1.5 or ≤2/3 and a *p* value of <0.05 were selected as DEPs. According to the standard, 595 DEPs were obtained. The volcano plot is a versatile figure. The horizontal axis represents the fold change of the probe, and the vertical axis represents the significance of the probe (−log10 *p* value). Figure 2 shows the differential expression patterns of proteins in the three groups (S24–S0, S60–S0, and S96–S0). As shown in Figure 3, the DEPs increased with storage time. In the S24–S0 group, 54 up-regulated proteins (Figure 3C) and 170 down-regulated proteins (Figure 3B) were identified for a total of 224 DEPs (Figure 3A). In the S60–S0 group, 160 up-regulated proteins (Figure 3C) and 302 down-regulated proteins (Figure 3C) were detected for a total of 462 DEPs (Figure 3A). A total of 207 up-regulated proteins and 356 down-regulated proteins (Figure 3B) were identified in the S96–S0 group for a total of 563 DEPs (Figure 3A).

### 3.3. Hierarchical Clustering Analysis

The hierarchical clustering analysis of DEPs more clearly shows the changes in protein abundance between the four time points (Figure 4). The colors of S0, S24, S60, and S96 changed significantly, indicating that the protein expression level changed substantially at the four time points. Significant changes in color between the S0, S24, and S60 groups indicated significant changes in protein expression.

### 3.4. GO Analysis of DEPs

According to the results for the DEPs identified in samples collected at different storage times, the numbers and percentages of differentially classified proteins in S24–S0, S60–S0, and S96–S0 are displayed in pie charts. Proteins were classified into the biological process (BP), cellular component (CC), and molecular function (MF) categories. The GO results showed that the main biological processes (Figure 5 (A-BP)) related to the DEPs in the S24–S0 group included cellular processes (180, 19%), metabolic processes (159, 17%), single-organism processes (146, 15%), cellular component organization or biogenesis (105, 11%), biological regulation (88, 9%), localization (64, 7%), etc. The DEPs in the S60–S0 group participated in several main biological processes (Figure 5 (B-BP)), including cellular process (373, 20%), metabolic process (337, 18%), single-organism process (292, 16%), cellular component organization or biogenesis (196, 10%), biological regulation (163, 9%), localization (127, 7%), response to stimulus (106, 6%), negative regulation of biological processes (65, 3%), etc. The DEPs in the S96–S0 group participated in several main biological processes (Figure 5 (C-BP)), including cellular processes (443, 20%), metabolic processes (397, 18%), single-organism processes (351, 16%), cellular component organization or biogenesis (230, 10%), biological regulation (180, 8%), localization (151, 7%), response to stimulus (128, 6%), negative regulation of biological processes (71, 3%), etc. Thus, cellular processes, metabolic processes, single-organism processes, cellular component organization or biogenesis, biological regulation, localization, response to stimulus, negative regulation of biological processes, positive regulation of biological processes, and other processes have important effects on the aging of *V. volvacea* fruiting bodies during storage at 15 °C. The main cellular components (Figure 5 (A-CC)) in which DEPs in the S24–S0 group were involved included cell parts (187, 26%), organelles (159, 22%), organelle parts (113, 16%), macromolecular complexes (99, 14%), the membrane (61, 9%), etc. The main cellular components (Figure 5 (B-CC)) for DEPs in the S60–S0 group included the cell parts (380, 27%), organelles (316, 22%), organelle parts (232, 16%), macromolecular complexes (205, 14%), the membrane (117, 8%), membrane parts (90, 6%), the membrane-enclosed lumen (83, 6%), etc. The main cellular components (Figure 5 (C-CC)) for DEPs in the S96–S0 group included cell parts (446, 27%), organelles (361, 21%), organelle parts (267, 16%), macromolecular complexes (237, 14%), the membrane (139, 8%), membrane parts (110, 7%), the membrane-enclosed lumen (96, 6%), etc. The number of DEPs increased with prolonged storage time. The main molecular functions (Figure 5 (A-MF)) of DEPs in the S24–S0 group included binding (158, 49%), catalytic activity (86, 27%), structural molecule activity (24, 7%), molecular function regulation (17, 5%), transporter activity (17, 5%), etc. The main molecular functions (Figure 5 (B-MF)) of DEPs in the S60–S0 group included binding (306, 46%), catalytic activity (203, 30%), structural molecule activity (51, 8%), transporter activity (37, 6%), molecular function regulation (30, 5%), nucleic acid-binding transcription factor activity (14, 2%), etc. The main molecular functions (Figure 5 (C-MF)) of DEPs in the S90–S0 group included binding (363, 46%), catalytic activity (253, 32%), structural molecule activity (53, 7%), transporter activity (43, 5%), molecular function regulation (31, 4%), nucleic acid-binding transcription factor activity (14, 2%), etc.

In general, compared with the proteins identified in the 0 h samples, the contents of DEPs involved in biological processes, cell components, and molecular functions gradually increased with prolonged storage time. More specifically, the contents of DEPs classified into cellular processes, metabolic processes, single-organism processes, cellular component organization or biogenesis, biological regulation, cell parts, organelles, organelle parts, macromolecular complexes, binding, catalytic activity, structural molecule activity, transporter activity, etc. increased with storage time. Biological regulation and other processes exerted important effects. The molecular functions of the DEPs mainly included catalytic activity and binding.

### 3.5. KEGG Pathway Analysis

A KEGG analysis of 595 DEPs was subsequently conducted to explore the metabolic pathways and functions of different proteins. The results revealed 88 different metabolic pathways in which S24–S0 DEPs were involved, 9 of which were significantly altered. One hundred and one different metabolic pathways in which S60–S0 DEPs were involved were identified, nine of which were significantly altered. DEPs in S96–S0 involved 111 different metabolic pathways, 13 of which were significantly altered. The DEPs in S24–S0, S60–S0, and S96–S0 were all involved in RNA transport, alpha-linolenic acid metabolism, and phenylalanine, tyrosine, and tryptophan biosynthesis (*p* < 0.05) (Figure 6). Moreover, the RNA transport pathway enriched the most DEPs in all pathways. In S60–S0 and S96–S0, the DEPs were involved in the pathways of fatty acid degradation, peroxisome, lysine biosynthesis, ribosome, and lysine degradation (*p* < 0.05), and more DEPs were enriched in the ribosome and peroxisome pathways than in the other pathways. However, only one protein was involved in the lysine biosynthesis and degradation pathways. Therefore, the related DEPs in the RNA transport, ribosomes, peroxisomes, lysine biosynthesis, and degradation pathways are the focus of further discussion.

### 3.6. Analysis of Protein–Protein Interactions (PPIs) among DEPs

The PPI network was analyzed in this study (Figure 7). The results revealed many similarities in the KEGG pathways between the S60–S0 (Figure 7B) and S96–S0 (Figure 7C) groups, both of which contained DEPs involved in the alpha-linolenic acid metabolism; phenylalanine, tyrosine, and tryptophan biosynthesis; RNA transport; fatty acid degradation; peroxisome; and ribosome pathways. In contrast, the S24–S0 (Figure 7A) group was significantly different from the S60–S0 and S96–S0 groups. Only the alpha-linolenic acid metabolism; phenylalanine, tyrosine, and tryptophan biosynthesis; and RNA transport pathways of these groups overlapped.

## 4. Discussion

### 4.1. Physiological Responses of V. volvacea to Suitable-Temperature Storage Conditions

Generally, the higher the storage temperature is, the greater the evaporation of water. Texture quality is also related to the water content [27]. When water loss reaches a certain level, the appearance of fruits and vegetables becomes withered, which may be the main reason for the increased hardness and fracturability of the fruiting bodies of *V. volvacea* observed in this experiment. Water evaporation causes the wilting of fruiting bodies, which is also the main reason for the increase in the WLR. Changes in REC and MDA reflect important signs of membrane system aging and damage [28,29]. In this experiment, with a prolonged storage time, the REC and MDA of *V. volvacea* increased, indicating that the cell membrane of *V. volvacea* was damaged during the aging process.

### 4.2. DEPs in V. volvacea Stored at 15 °C after Harvest

The proteomic analysis of *V. volvacea* was conducted after 0, 24, 60, and 96 h of storage using iTRAQ-based proteomics technology. Carbohydrate and energy metabolism, amino acid biosynthesis and metabolism, signal transduction, and other pathways (Figure 8) related to *V. volvacea* storage were screened using GO, KEGG, and PPI analyses. At the same time, the effects of DEPs, such as Polyadenylate-binding protein 1 (PAB1), Eukaryotic translation initiation factor 3 subunit A 1 (RPG1), Acetyl-CoA carboxylase 1 (ACC1), Alcohol dehydrogenase 3 (ADH3), Alcohol dehydrogenase 2 (ADH2), Aldehyde dehydrogenase 5 (ALD5), and Succinate dehydrogenase 2 (SDH2) (Figure 8) involved in pathways for the fruiting-body aging of *V. volvacea* during storage at 15 °C were discussed.

#### 4.2.1. DEPs Associated with RNA Transport

The transfer of RNA from the nucleus to the cytoplasm is the basis of gene expression. The RNA transport, ribosome, and mRNA surveillance pathways, and other pathways related to transcription were involved in *V. volvacea* aging during storage at 15 °C.

*V. volvacea* is a high-temperature edible fungus with a short storage period. The rate of senescence of *V. volvacea* is rapid after harvest. The importance of the fungal stress response at the molecular level reduces gene expression under unpredictable stress conditions, eukaryotic cells block the translation of mRNA during transcription, senescent cells are formed at this stage, and complex polymers of RNA and proteins in the cytoplasm are formed [30]. PAB1 protein production is induced by stress; namely, the accumulation of stress is determined by tracking the process of PAB1 gene expression and its RNA distribution under stress [30]. In this experiment, the content of PAB1 decreased significantly during storage. Thus, the ability of the fruiting bodies to combat stress may decrease during aging. In addition, RPG1 encodes a membrane integrin to maintain cell membrane integrity. RPG1 may also maintain membrane integrity by interacting with other membrane proteins and, in some way, coordinate the network of the primary outer wall [31]. RPG1 is reported to be sugar transporter 40; thus, it is speculated that RPG1 may be involved in the transport of extracellular matrix components or other precursor components [32]. In this study, the expression of the RPG1 protein decreased with prolonged storage time, potentially due to complete damage to the cell membrane. At the same time, the results of REC and MDA assays also showed that the cell membrane was damaged during storage.

In addition, studies have shown that, with increasing stress levels, *Saccharomyces cerevisiae* Hansen exhibits decreased stability of most of its own mRNA [33] and a reduced translation rate [34]. The expression of many ribosomal proteins is decreased at high temperatures. The decrease in the expression of many ribosomal proteins at high temperatures indicates that the biogenesis of ribosomes in *Spirulina* is inhibited. In this experiment, all of the ribosome-related DEPs were down-regulated. Ribosomal proteins not only synthesize new ribosomes with ribosomal RNA, but also function in vitro to regulate gene transcription and cell proliferation, differentiation, and apoptosis. Therefore, ribosomal function decreased during the aging process of *V. volvacea*, which may have led to a decrease in related gene regulation.

#### 4.2.2. DEPs Associated with Fatty Acid Biosynthesis and Metabolism

Lipids play an important role in many biological processes of plants, such as the closure of cells and organelles, the mediation of information exchange networks, the protection of tissues from harsh environments, the storage of necessary energy, signaling, and participation in photosynthetic capture [33]. In this research, the results were related to the synthesis and metabolism of fatty acids through alpha-linolenic acid metabolism, ether lipid metabolism, fatty acid degradation, fatty acid metabolism, and linoleic acid metabolism.

Changes in the fatty acid composition affect the cell membrane, which may accelerate the ripening and senescence of fruits. During the ripening and senescence of kiwi fruits, linoleic acid and linolenic acid significantly promote the accumulation of C6 aldehydes, and similar results have been reported in strawberry fruits. Jasmonate (JA) is a type of stress regulator that transduces signals in plants in response to environmental stress (especially salt stress). According to previous research reports, the exogenous application of JAs has been shown to be effective at improving the salt tolerance of many plants. ACCase is the key enzyme or rate-limiting enzyme involved in fatty acid biosynthesis, and fatty acids are the main raw material for triacylglycerol (TAG) synthesis [35]. Li and his colleagues have shown that the expression of the ACCase gene promotes fatty acid biosynthesis [36]. ACCase exerts a feedback-inhibitory effect on fatty acids in the plastid of plant cells. The accumulation of 18:1acp in the plastid directly inhibits the production of ACCase, resulting in a decrease in fatty acid synthesis [37]. ACCase has three functions: it is a biotin carboxyl carrier protein, biotin carboxylase, and carboxyltransferase. It is involved in the synthesis of long-chain fatty acids, which are essential for maintaining the function of the nuclear membrane. A positive correlation has been observed between the relative expression of the BN ACC1 gene and the accumulation of oil in the mature stage of seeds because of the extensive metabolism of oil, and the content of fatty acids increases rapidly [38]. In this study, the expression of ACC1 was significantly up-regulated. Fatty acid metabolism might increase during storage. According to previous reports, changes in fatty acid composition affected the cell membrane, which may accelerate the ripening and senescence of *V. volvacea* fruiting bodies.

In addition, ADH3 was up-regulated during storage for 60 h. ADH is a common enzyme expressed in plant tissues, and its relationship with plant stress resistance has been confirmed by the academic community. Huang et al. observed that ADH was related to the low-temperature tolerance of rice [39]. ADH maintains a higher energy level, protects the integrity of membrane structure and function, and prolongs the life span of plant cells by enhancing the Pasteur effect [40]. The increase in ADH3 expression in salt-tolerant rice is an adaptive response to salt stress. The excess production, accumulation, and imbalance of reactive oxygen species in rice cells may be caused by salt stress, and reactive oxygen species may lead to membrane peroxidation and degreasing, resulting in the destruction of the cell structure and function [41]. It has been reported that the relative expression of ADH in cotton roots changes under flooding stress [42]. Under short-term flooding and a hypoxic environment, the expression of ADH continues to rise in cotton. Ethanol fermentation is an adaptive mechanism for cotton to resist hypoxic stress [42]. In addition, flooding stress had a significant effect on the expression of the ADH2 gene in rice [43]. In this study, this result showed that ADH2 and ADH3 proteins were up-regulated with 60 h and 96 h of storage, which may have been an adaptive response of the straw mushroom fruiting body during senescence.

#### 4.2.3. DEPs Associated with Amino Acid Biosynthesis and Metabolism

Amino acid biosynthesis and metabolism are very important in development and the stress response [44,45,46,47]. Amino acids function as building blocks and provide energy for fungal growth and development. They are major long-distance nitrogen transport carriers in fungal systems, important metabolites, molecular forms of nitrogen storage, stress response signal-transduction molecules, and molecular precursors [17]. The balance between protein synthesis and degradation plays a crucial role in regulating biological cell processes and their responses to developmental or environmental cues [48,49]. Fungal senescence involves many dynamic changes in proteins [50]. In addition, little is known about the change in the protein expression of *V. volvacea* under suitable storage conditions, including a suitable temperature. In this experiment, the results showed that the DEPs were enriched in phenylalanine, tyrosine, and tryptophan biosynthesis; valine, leucine, and isoleucine degradation; and lysine biosynthesis and degradation. ALD5 is involved in maintaining mitochondrial aldehyde dehydrogenase on the electron transport chain in *S. cerevisiae*. The functional analysis of the ALD gene family in *S. cerevisiae* during anaerobic fungal growth on glucose showed that the NADP+-dependent ALD6 and ALD5 isoforms played a major role in acetate formation [51]. Blomberg and Adler found that acetate production was enhanced during osmotic stress [52]. ALD2 and ALD6 are known to be induced by osmotic stress [53]. ALD5 seems to respond to osmotic stress from the point of view of mitochondrial respiration as well [54]. In this experiment, compared with 0 h, ALD6 and ALD5 were significantly up-regulated and involved in amino acid biosynthesis and metabolism, which may play important roles in the osmotic stress induced by fruiting bodies’ senescence.

#### 4.2.4. DEPs Associated with Other Biological Processes

In addition, DEPs such as SDH2 were up-regulated at 60 h and 96 h during storage. Succinate dehydrogenase (SDH) is the key enzyme involved in the tricarboxylic acid cycle [55]. SDH can be used as an index for evaluating the levels of the tricarboxylic acid cycle [56], changes in which can reflect the functional state of mitochondria, and is an important respiratory enzyme [57]. Few studies have investigated SDH2 under stress, but its expression affects energy production. In this study, the SDH2 protein was up-regulated with 60 h and 96 h storage, which is consistent with the previous research results of low-temperature environmental storage [5]. The up-regulation of the SDH2 protein may provide more energy for *V. volvacea* cells to cope with the aging process.

## 5. Conclusions

In conclusion, the data obtained in this study may help to describe the mechanism of postharvest senescence of *V. volvacea* stored at 15 °C. This mechanism may be related to the regulation of DEPs involved in protein synthesis and transcription, fatty acid synthesis and metabolism, and amino acid synthesis and metabolism. In addition, most RNA transport-related proteins were down-regulated, which may lead to a decrease in related gene regulation. The expression of other important proteins, such as the fatty acid metabolism related proteins increased; and changes in fatty acid composition affected the cell membrane, which may accelerate the ripening and perception of *V. volvacea* fruiting bodies. The study will provide a new perspective for studying the molecular mechanism of *V. volvacea* during postharvest storage at 15 °C.

## Figures and Tables

**Figure 1 jof-08-00819-f001:**
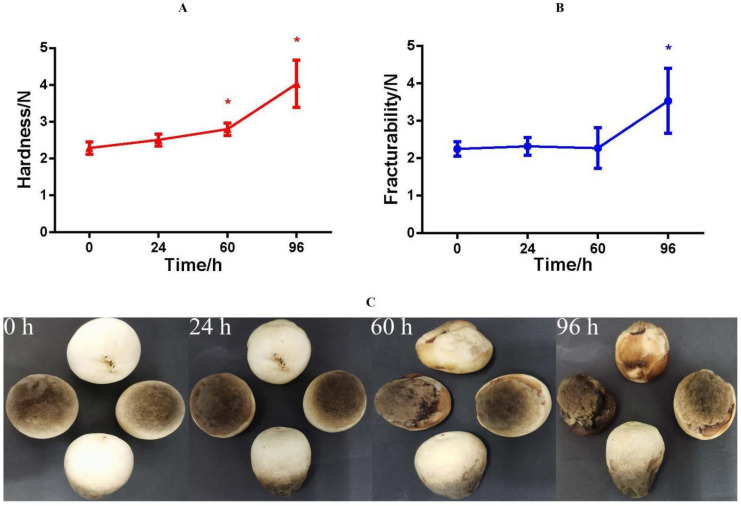
Changes in the hardness (**A**), fracturability (**B**), and phenotype (**C**) of *V. volvacea* fruiting bodies during different storage periods. Each bar represents the mean ± SE of the individual samples, and * represents *p* < 0.05.

**Figure 2 jof-08-00819-f002:**
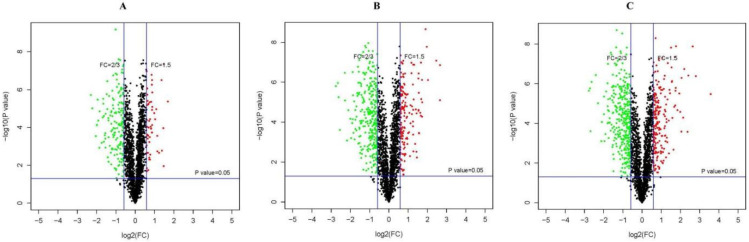
(**A**–**C**) Volcano plots showing DEPs in *V. volvacea* S24–S0, S60–S0, and S96–S0. Dots highlighted in red (FC ≥ 1.5) and green (FC ≤ 2/3) indicate proteins that exhibited significantly different expression levels (*p* ≤ 0.05).

**Figure 3 jof-08-00819-f003:**
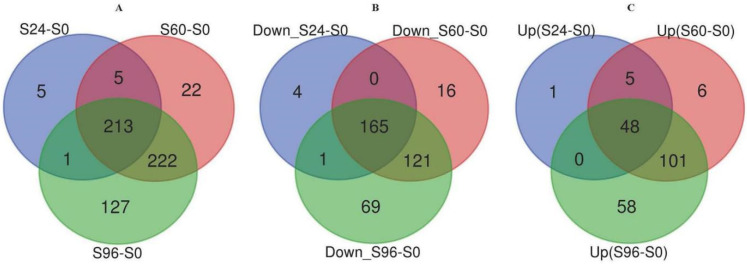
(**A**–**C**) Venn diagrams showing the overlapping DEPs between the three *V. volvacea* groups (S24–S0, S60–S0, and S96–S0). The numbers of DEPs that were down-regulated (**B**) and up-regulated (**C**) at each developmental stage are shown in different circles.

**Figure 4 jof-08-00819-f004:**
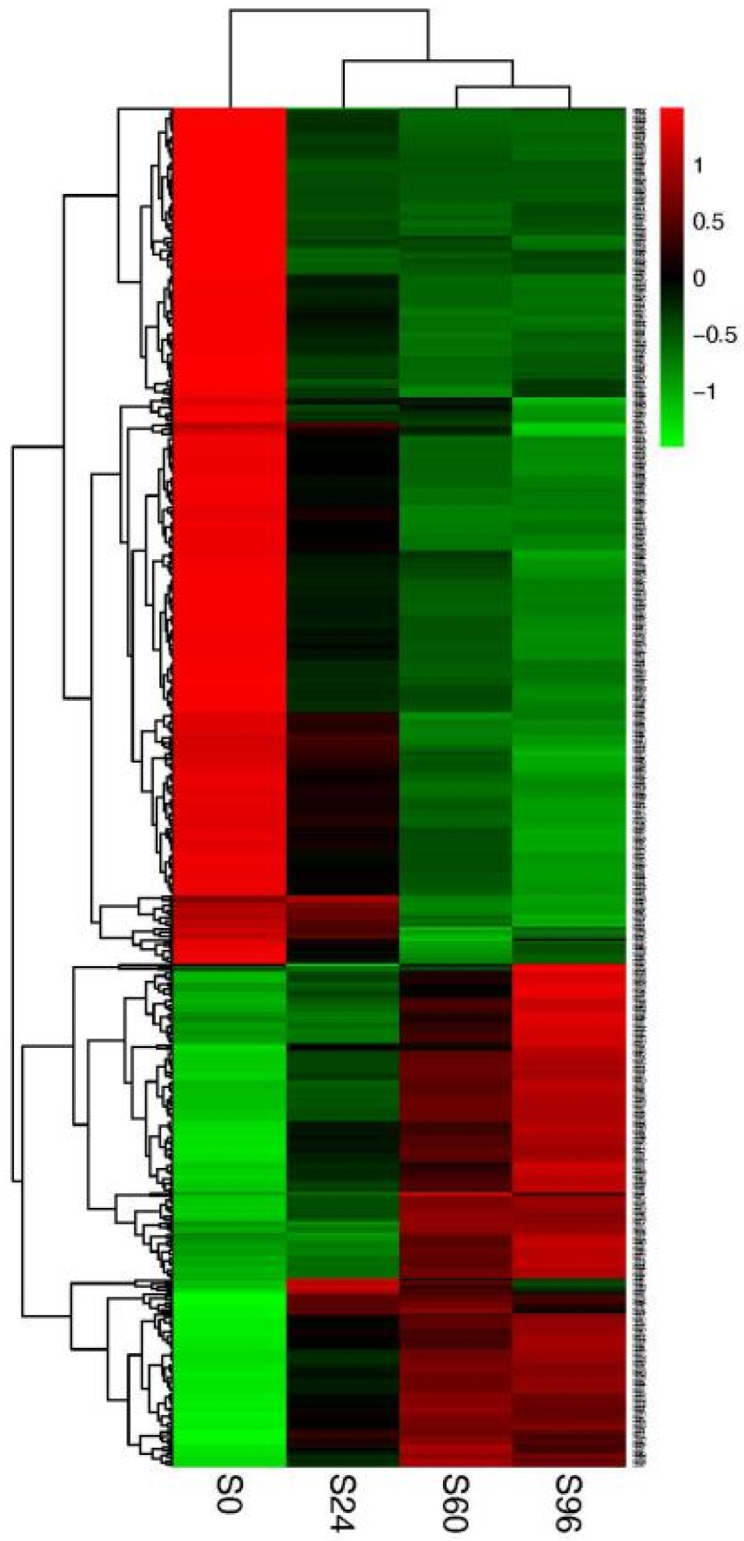
Clustering analysis of *V. volvacea* DEPs in the S24, S60, and S96 groups compared with the S0 group. Red, green, and black indicate an increase, decrease, and no change in protein abundance, respectively, compared with the baseline level.

**Figure 5 jof-08-00819-f005:**
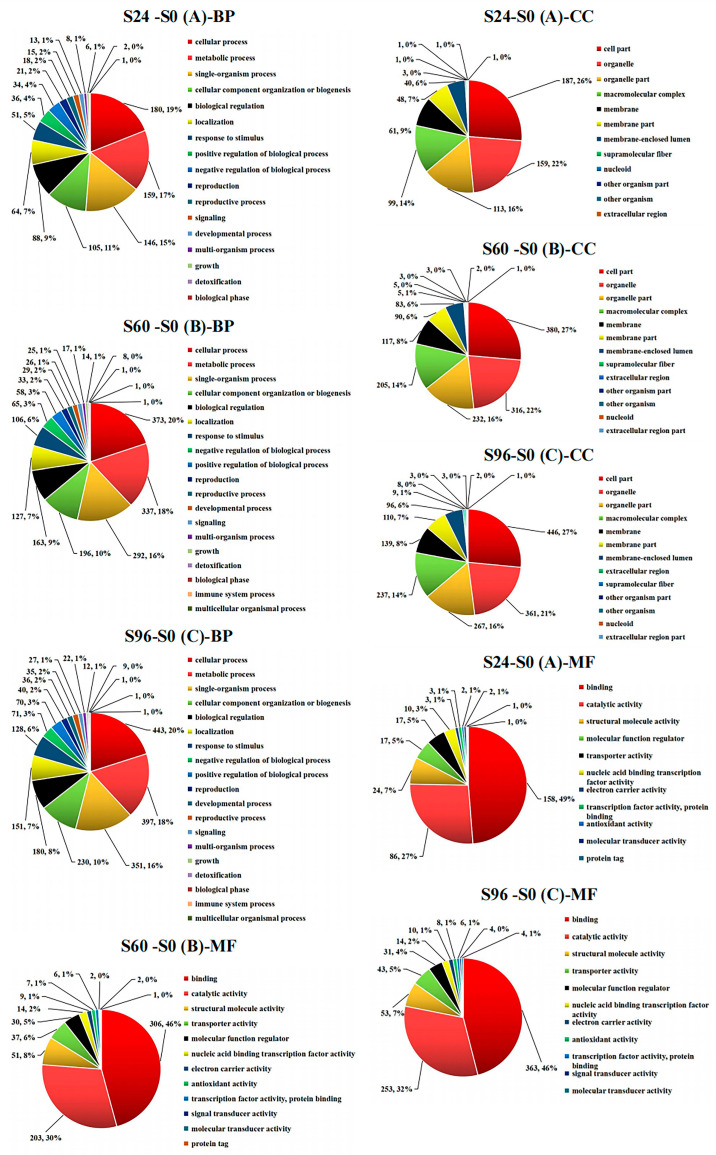
Bioinformatics analysis of GO terms for the aforementioned DEPs in three domains: BP, MF, and CC. The statistics at GO level 2 are shown in this figure. A: S24–S0; B: S60–S0; and C: S96–S0.

**Figure 6 jof-08-00819-f006:**
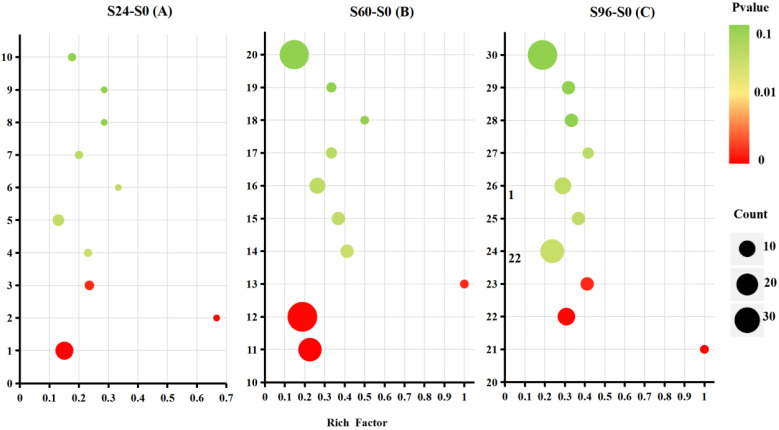
Bubble diagrams of the first 10 enriched KEGG pathways in S24–S0 (**A**), S60–S0 (**B**), and S96–S0 (**C**). Different numbers on the Y axis represent the following KEGG pathways: 1, RNA transport; 2, alpha-linolenic acid metabolism; 3, phenylalanine, tyrosine, and tryptophan biosynthesis; 4, valine, leucine, and isoleucine degradation; 5, mRNA surveillance; 6, ABC transporters; 7, pentose and glucuronate interconversions; 8, ether lipid metabolism; 9, nitrogen metabolism; 10, steroid biosynthesis; 11, RNA transport; 12, ribosome; 13, alpha-linolenic acid metabolism; 14, phenylalanine, tyrosine, and tryptophan biosynthesis; 15, fatty acid degradation; 16, peroxisome; 17, lysine degradation; 18, ABC transporters; 19, lysine biosynthesis; 20, biosynthesis of antibiotics; 21, alpha-linolenic acid metabolism 22, pyruvate metabolism; 23, phenylalanine, tyrosine, and tryptophan biosynthesis; 24, RNA transport; 25, fatty acid degradation; 26, peroxisome; 27, lysine biosynthesis; 28, SNARE interactions in vesicular transport; 29, fatty acid metabolism; and 30, ribosome. The rich factor represents the ratio between the DEPs and all annotated proteins enriched in the pathway. The rich factor refers to the ratio of the number of differentially expressed proteins located in the go entry to the total number of transcripts located in the go entry in all annotated proteins. The greater the rich factor, the higher the degree of enrichment. The bubble scale represents the number of DEPs, and the depth of the bubble color represents the adjusted *p* value.

**Figure 7 jof-08-00819-f007:**
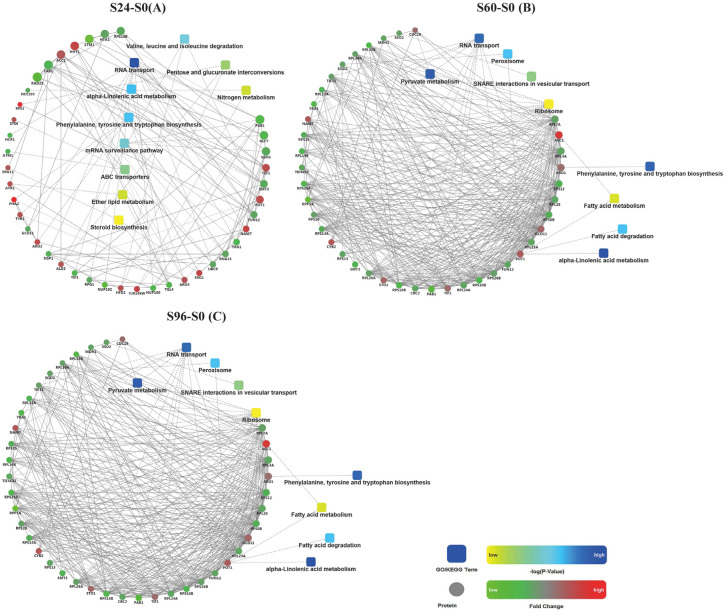
PPI network diagram. (**A**): S24–S0; (**B**): S60–S0; and (**C**): S96–S0. Note: Dots represent proteins, red represents up-regulated expression, and green represents down-regulated expression. Rounded rectangles represent biological processes, cell localization, molecular functions, or signaling pathways; blue represents high significance, and yellow represents low significance. Straight lines represent the interaction relationship, the solid lines represent the relevant relationships verified in this report, and the dashed lines represent relationships unconfirmed by this experiment.

**Figure 8 jof-08-00819-f008:**
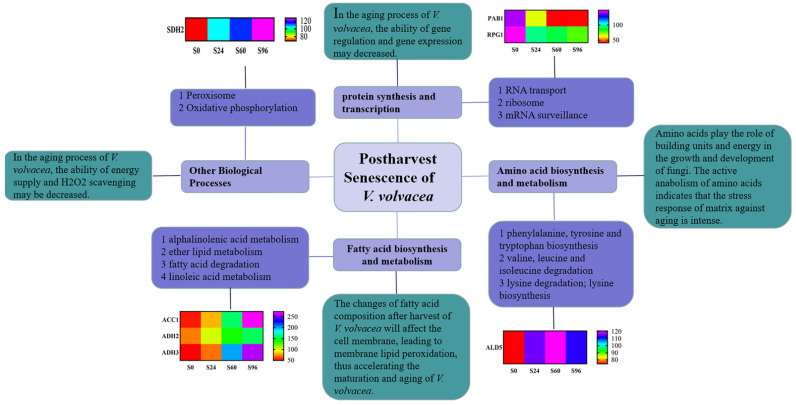
Model of *V. volvacea* postharvest senescence based on physiological, biochemical, and proteomic changes.

## Data Availability

Not applicable.

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
