# Peer review of "Comparative Proteomics Study on the Postharvest Senescence of Volvariella volvacea"

_jof, 2022, doi:10.3390/jof8080819_

Round 1
Reviewer 1 Report
Volvariella volvacea is a high-temperature edible fungus, which is very delicious and nutritious. However, this mushroom has a short storage period after harvest. The molecular mechanism underlying the postharvest senescence of V. volvacea stored at 15 °C still remains unclear, which is crucial to the shelf period and circulation of fresh fruiting bodies.
Here is some suggestions to improve the manuscript:
1) Line 17-18, "Four time points (0, 24, 60 and 96 h) were selected for the comparative proteomics study of V. volvacea after storage at 15 °C." "after" changed to "during" would be better.
2) Line 27, delete "storage".
3) Line 103-104, Line 111, please check the "Supplementary Material".
4) Line 126, "The texture results showed that hardness is significant during 60-96 h (Figure 1A)", please check the tense when results of this research referred to.
5) Line 134, change "Figure 1. (A, B and D)" to "Figure 1. (A, B and C)".
6) Line 202, "organelle (361, 21%)" not "organelle (316, 21%)".
7) Line 232, change "S96-S0 DEPs" to " DEPs in S96-S0".
8) Line 295, check the meaning of "Carnitine O-acetyltransferase 2", and correct related content.
9) The form of references should be modified, such as 31, Line 526.
Author Response
Response to Reviewer 1 Comments
Dear Reviewers and Editor,
First, I want to thank you for reviewing our article. Your comments were very professional, and we appreciate all the noted weaknesses of our manuscript. We believe that the revised article will be more suitable after revision. Modifications have been made to the original text base on the comments provided by the reviewers. The reviewers’ comments and the responses are as follows:
Point 1: Line 17-18, "Four time points (0, 24, 60 and 96 h) were selected for the comparative proteomics study of V. volvacea after storage at 15 °C." "after" changed to "during" would be better.
Response 1: Thank you for your kind suggestion. The sentences have been rewritten as follows: Four time points (0, 24, 60 and 96 h) were selected for the comparative proteomics study of V. volvacea during storage at 15 ℃.
Point 2: Line 27, delete "storage".
Response 2: Line 27, "storage" has been deleted .
Point 3: Line 103-104, Line 111, please check the "Supplementary Material".
Response 3: We revised “Supplementary Material” to “The methods are described for Zha et al [5].”
Point 4: Line 126, "The texture results showed that hardness is significant during 60-96 h (Figure 1A)", please check the tense when results of this research referred to.
Response 4: We revised “The texture results showed that hardness is significant during 60-96 h (Figure 1A)” to “The texture results showed that hardness was significant during 60-96 h (Figure 1A)”
Point 4: Line 126, "The texture results showed that hardness is significant during 60-96 h (Figure 1A)", please check the tense when results of this research referred to.
Response 4: We revised “The texture results showed that hardness is significant during 60-96 h (Figure 1A)” to “The texture results showed that hardness was significant during 60-96 h (Figure 1A)”
Point 5: Line 134, change "Figure 1. (A, B and D)" to "Figure 1. (A, B and C)".
Response 5: We revised “Figure 1. (A, B and D)” to “Figure 1. (A, B and C)”
Point 6: Line 202, "organelle (361, 21%)" not "organelle (316, 21%)".
Response 6: We revised “organelle (361, 21%)” to “organelle (316, 21%)”
Point 7: Line 232, change "S96-S0 DEPs" to " DEPs in S96-S0".
Response 7: We revised “S96-S0 DEPs involved 111 different metabolic pathways, 13 of which were significantly altered. to “DEPs in S96-S0 involved 111 different metabolic pathways, 13 of which were significantly altered.”
Point 8: Line 295, check the meaning of "Carnitine O-acetyltransferase 2", and correct related content.
Response 8: “Carnitine O-acetyltransferase 2” has been deleted.
Point 9: The form of references should be modified, such as 31, Line 526.
Response 9: We revised “Guan, Y.-F.; Huang, X.-Y.; Zhu, J.; Gao, J.-F.; Zhang, H.-X.; Yang, Z.-N. RUPTURED POLLEN GRAIN1, a member of the MtN3/saliva gene family, is crucial for exine pattern formation and cell integrity of microspores in Arabidopsis. Plant Physiol. 2008. 147, 852-863. https://doi.org/10.1104/pp.108.118026” to “Guan, Y.F.; Huang, X.Y.; Zhu, J.; Gao, J.F.; Zhang, H.X.; Yang, Z.N. RUPTURED POLLEN GRAIN1, a member of the MtN3/saliva gene family, is crucial for exine pattern formation and cell integrity of microspores in Arabidopsis. Plant Physiol. 2008. 147, 852-863. https://doi.org/10.1104/pp.108.118026”
Reviewer 2 Report
Review of Comparative Proteomics Study on the Postharvest Senescence 2 of Volvariella volvacea at 15 ℃ by Zha et al.
This paper is a very thorough investigation of proteins in V. volvacea during 4 days of 15C storage using iTRAQ labelling and 2-DE at 4 time points, from time 0 to 96 h. I would like to see an explanation in the introduction as to why these mushrooms must be stored at 15C and not close to 0C which is the normal mushroom storage. Are there some proteolytic enzymes that are activated by a low temperature storage, or some other biochemical reason that they cannot be held at a low temperature?
In their experiments, the authors identified over 2000 proteins of which 595 were DEPs (differentially expressed) and 213 were common to all three times at 15C. The authors analysed their data with GO (Gene Ontology), KEGG (Kyoto Encyclopedia of Genes and Genomes), and PPI (protein-protein interactions). Figure 4 shows hierarchical clustering analysis, which I think should be revised. They are comparing the time points of S24, S60 and S96 to S0, and S0 should be black since it is the reference.
At the end of the KEGG analysis on ln 234 the authors say that the DEPs of RNA transport, ribosome, peroxisome, lysine biosynthesis and degradation were the focus of further discussion. I did not see where these functions were explicitly discussed further, and would remove this sentence.
The discussion is the weak point of the paper.
In the discussion they chose 8 DEPs, each of which representative of a particular biochemical pathway to discuss the whole pathway. One DEP is CAT2, which is identified on ln 288 as carnitine O-acetyltransferase 2 and on ln 409 as catalase and discussed as such. Some of the chosen DEPs increase during the mushroom senescence and some decrease. Many of them are involved in more than one biochemical process although only one is discussed here.
For RNA transport the authors examined polyadenylate-binding protein 1 (PAB1) and eukaryotic translation initiation factor 3 subunit A 1 (RPG1), both of which decreased during the 96 h at 15C. the authors say that PAB1 is induced by stress, but here it decreased even at the 24 h time point. The eukaryotic translation factor 3 subunit A1 is said by the authors to encode a membrane integrin to maintain cell membrane integrity. I could not find that function at all on the literature. RPG1 might be a membrane protein but not what it is identified as in this paper. In addition, in this section of the discussion they authors have placed figure 9 which is very difficult to read but appears to be pathways in RNA transport. There is no discussion on this in the text and the figure legend does not explain the different colors around the various proteins. What are white and green?
In the discussion on membranes with the DEP Acetyl-CoA carboxylase 1 (ACC1) which increased greatly during the 96 h, there is along section of kiwifruit ripening and jasmonic acid, with no references and limited relevance to the subject at hand. Google says that ACC1 is a multifunctional enzyme that catalyzes the carboxylation of acetyl-CoA, forming malonyl-CoA, which is used in the plastid for fatty acid synthesis and in the cytosol in various biosynthetic pathways including fatty acid elongation. The authors do not explain why, if the mushrooms are undergoing senescence there would be significant increase in fatty acid synthesis.
I think that the paper has a lot of interesting data, not to mention a LOT of data, but it may have overwhelmed the authors in how to discuss it coherently and without a revision of the discussion, I do not recommend publication.
Author Response
Response to Reviewer 2 Comments
Dear Reviewers and Editor,
First, I want to thank you for reviewing our article. Your comments were very professional, and we appreciate all the noted weaknesses of our manuscript. We believe that the revised article will be more suitable after revision. Modifications have been made to the original text base on the comments provided by the reviewers. The reviewers’ comments and the responses are as follows:
Point 1: This paper is a very thorough investigation of proteins in V. volvacea during 4 days of 15 ℃ storage using iTRAQ labelling and 2-DE at 4 time points, from time 0 to 96 h. I would like to see an explanation in the introduction as to why these mushrooms must be stored at 15℃ and not close to 0℃ which is the normal mushroom storage. Are there some proteolytic enzymes that are activated by a low temperature storage, or some other biochemical reason that they cannot be held at a low temperature?
Response 1: Your comments were very professional. The storage of V. volvacea below 10 ℃ easily leads to autolysis of the fruiting bodies [6, 7]. Alternatively, the storage of fresh V. volvacea at 25 ℃ increases its respiratory rate and metabolic activity, and the quality of fresh V. volvacea fruiting bodies decreases rapidly within 48 h of storage, thus losing the commercial value. Meanwhile, researchers found that the best storage of V. volvacea was achieved when the temperature was approximately 15 ℃[8]. However, At present, we know this phenomenon. The internal mechanism of straw mushroom autolysis at low temperature is not clear.
Point 2: In their experiments, the authors identified over 2000 proteins of which 595 were DEPs (differentially expressed) and 213 were common to all three times at 15 ℃. The authors analysed their data with GO (Gene Ontology), KEGG (Kyoto Encyclopedia of Genes and Genomes), and PPI (protein-protein interactions). Figure 4 shows hierarchical clustering analysis, which I think should be revised. They are comparing the time points of S24, S60 and S96 to S0, and S0 should be black since it is the reference.
Response 2: Your suggestion is very good. However, the software defaults to this form and cannot be changed. I'm very sorry
Point 3: At the end of the KEGG analysis on ln 234 the authors say that the DEPs of RNA transport, ribosome, peroxisome, lysine biosynthesis and degradation were the focus of further discussion. I did not see where these functions were explicitly discussed further, and would remove this sentence.
Response 3: We revised “Therefore, the RNA transport, ribosome, peroxisome, lysine biosynthesis and degradation pathways and related DEPs were the focus of further discussion.” to “Therefore, the related DEPs in RNA transport, ribosomes, peroxisomes, lysine biosynthesis and degradation pathways are the focus of further discussion.”
Point 4: In the discussion they chose 8 DEPs, each of which representative of a particular biochemical pathway to discuss the whole pathway. One DEP is CAT2, which is identified on ln 288 as carnitine O-acetyltransferase 2 and on ln 409 as catalase and discussed as such. Some of the chosen DEPs increase during the mushroom senescence and some decrease. Many of them are involved in more than one biochemical process although only one is discussed here.
Response 4: Thank you for your reminder. I have realized that this is an error and deleted the description of CAT2.
Point 5: For RNA transport the authors examined polyadenylate-binding protein 1 (PAB1) and eukaryotic translation initiation factor 3 subunit A 1 (RPG1), both of which decreased during the 96 h at 15C. the authors say that PAB1 is induced by stress, but here it decreased even at the 24 h time point. The eukaryotic translation factor 3 subunit A1 is said by the authors to encode a membrane integrin to maintain cell membrane integrity. I could not find that function at all on the literature. RPG1 might be a membrane protein but not what it is identified as in this paper. In addition, in this section of the discussion they authors have placed figure 9 which is very difficult to read but appears to be pathways in RNA transport. There is no discussion on this in the text and the figure legend does not explain the different colors around the various proteins. What are white and green?
Response 5: Thank you for your reminder. The descriptions of PAB1 (Yang, X, X. Related genes and regularity effect stress granules formation in Saccharomyces cerevisiae Hansen. Ph.D, Harbin Institute of technology, Harbin, China. 2015. https://kns.cnki.net/KCMS/detail/detail.aspx?dbname=CDFDLAST2016&filename=1015957481.nh ) and RPG1 (Guan, Y.F.; Huang, X.Y.; Zhu, J.; Gao, J.F.; Zhang, H.X.; Yang, Z.N. RUPTURED POLLEN GRAIN1. RUPTURED POLLEN GRAIN1, a member of the MtN3/saliva gene family, is crucial for exine pattern formation and cell integrity of microspores in Arabidopsis. Plant Physiol. 2008. 147, 852-863. https://doi.org/10.1104/pp.108.118026) have similar expressions in the literature. We add a description of the discussion about figure 9 to the original text. At the same time, the description of the figure is also added in the figure annotation about white and green.
Point 6: In the discussion on membranes with the DEP Acetyl-CoA carboxylase 1 (ACC1) which increased greatly during the 96 h, there is along section of kiwifruit ripening and jasmonic acid, with no references and limited relevance to the subject at hand. Google says that ACC1 is a multifunctional enzyme that catalyzes the carboxylation of acetyl-CoA, forming malonyl-CoA, which is used in the plastid for fatty acid synthesis and in the cytosol in various biosynthetic pathways including fatty acid elongation. The authors do not explain why, if the mushrooms are undergoing senescence there would be significant increase in fatty acid synthesis.
Response 6: We revised “In this study, the expression of ACC1 was significantly up-regulated, indicating increased fatty acid metabolism in V. volvacea during storage.” to “In this study, the expression of ACC1 was significantly up-regulated. Fatty acid metabolism might increase during storage. According to previous reports, changes in fatty acid composition affect the cell membrane, which may accelerate the ripening and senescence of fruits.”
Reviewer 3 Report
The manuscript by Zha et al. describes a study of the postharvest senescence of Volvariella volvacea at 15 ℃, through a comparative proteomics approach. Although the subject may be of relevance the study and manuscript suffer from major problems.
First, and more importantly, the authors fail to explain convincingly how a comparative proteomics study may help to solve the problem of postharvest senescence of this mushroom. At the end, and after all the discussion, the main conclusion is still “the data obtained in this study may help describe the mechanism of postharvest senescence of V. volvacea stored at 15ºC”. So, the question is why the authors couldn’t do it (at least in part) in the current study.
The results present very generic processes (biological, cellular, molecular) and do not really highlight much related postharvest senescence. The discussion is poor, rather vague and a bit rambling. It clearly needs to the redone and focus on senescence mechanisms in fungi. Also, much of the discussion involves aspects of senescence in fruits and vegetables and fungal fruiting bodies are neither!
Finally, the language needs a good revision and edit for grammar and style.
Specific comments:
- the abstract lacks the main conclusions of this study.
- the introduction begins with senescence in fruit development, but fungal fruiting bodies are not truly fruits. So, does this make sense?
- lines 70- 79: This is somehow confusing. It is not clear what was done previously. This whole paragraph does not make much sense here. The final paragraph of the introduction should state very clearly which were the aims of the study.
- line 82: How many samples were analyzed in total? How many for each time? In fact, each sample corresponded to a single mushroom or were there composite samples? This should be clearly described.
It is not clear if replicates were used. They should have been at least 3.
- line 115-117: Very vague description. Please describe in more detail. In fact, there is nothing about statistics here.
- line 126: water production? is this correct? If so, what do you mean exactly?
- figure 5 is too small and impossible to read. It is almost useless like this.
- line 232-233: why is this an attractive result?
- line 280 (item 4.2): there is no discussion here!
- line 296-298: is this really surprising or unexpected?
- line 307: which protein is PAB1? function?
- line 310-311: again not really surprising or unexpected.
- figure 9 is too small, low quality and impossible to read. It is almost useless like this.
- line 336: not sure of what closure means.
- line 350-352: what does this has to do with fruit senescence?
Author Response
Response to Reviewer 3 Comments
Dear Reviewers and Editor,
First, I want to thank you for reviewing our article. Your comments were very professional, and we appreciate all the noted weaknesses of our manuscript. We believe that the revised article will be more suitable after revision. Modifications have been made to the original text base on the comments provided by the reviewers. The reviewers’ comments and the responses are as follows:
Point 1: the abstract lacks the main conclusions of this study.
Response 1: Thank you for your kind suggestion. The abstract have been rewritten as follows: V. volvacea is difficult to store after harvest, which restricts the production and circulation of V. volvacea fruiting bodies. Low-temperature storage is the traditional storage method used for most edible fungi. However, V. volvacea undergoes autolysis at low temperature. However, when fruiting bodies are stored at 15 ℃ (suitable temperature), V. volvacea has the best fresh keeping effect. However, the molecular mechanism underlying the postharvest senescence of V. volvacea remains unclear. Based on this information, we stored V. volvacea fruiting bodies at 15 ℃ after harvest and then analyzed the texture and phenotype combined with the results of previous physiological research. Four time points (0, 24, 60 and 96 h) were selected for the comparative proteomics study of V. volvacea during storage at 15 ℃. A variety of proteins showed differential expression in postharvest V. volvacea at 15 ℃. Further comparison of the gene ontology (GO) enrichment analysis and KEGG pathways performed at different sampling points revealed proteins that were significantly enriched at several time points. At the same time, we also analyzed differentially expressed proteins (DEPs) related to RNA transport, fatty acid biosynthesis and metabolism, and amino acid biosynthesis and metabolism pathways and discussed the molecular functions of the PAB1, RPG1, ACC1, ADH3, ADH2, ALD5, and SDH2) proteins in postharvest V. volvacea senescence. Our results showed that many biological processes of Postharvest Senescence of V. volvacea changed. Most importantly, we found that most RNA transport related proteins were down regulated. which may lead to a decrease in related gene regulation. Our results also showed that other important proteins, such as fatty acid metabolism protein, also increased, changes in fatty acid composition affect the cell membrane, which may accelerate the ripening and perception of fruits. Therefore, our research provides a reference for us to further study the aging mechanism of V. volvacea.
Point 2: the introduction begins with senescence in fruit development, but fungal fruiting bodies are not truly fruits. So, does this make sense?
Response 2: Although edible fungi are not real fruits, they also age after harvest. The changes of fruits after harvest are also applicable to edible fungi.
Point 3: lines 70- 79: This is somehow confusing. It is not clear what was done previously. This whole paragraph does not make much sense here. The final paragraph of the introduction should state very clearly which were the aims of the study.
Response 3: We have rewritten the paragraph as follow: Therefore, fruiting bodies of V. volvacea were selected to be stored at 15 ℃. Based on our previous study[22], in which the changes of morphological and physiological indexes at 0, 12, 24, 36, 48, 60, 72 and 96 h were measured; four time points, namely, 0, 24, 60 and 96 h, were selected for this comparative proteomics study utilizing iTRAQ technology. From the perspective of proteomics, the dynamic changes in the proteome during the postharvest storage of V. volvacea at 15 ℃ were revealed, which provided a theoretical basis for the development of enzyme inhibitors to inhibit the aging of V. volvacea., thereby prolonging the fresh-keeping period of this mushroom. At the same time, our results will provide a new perspective for studying the molecular mechanism of V. volvacea postharvest storage at 15 ℃.
Point 4: How many samples were analyzed in total? How many for each time? In fact, each sample corresponded to a single mushroom or were there composite samples? This should be clearly described.
Response 4: We have rewritten the paragraph as follow: Samples of V. volvacea V23 were collected from the Shanghai Fanshun Edible Fungi Professional Cooperative and immediately sent to the lab of Shanghai Academy of Agricultural Sciences for treatment. We selected full, smooth-surfaced and disease-free fruiting bodies, and the size of the complete fruiting bodies was approximately the same across samples. The samples were randomly divided into four groups (3 fruit bodies in each group), with 3 replicates in each group. The samples were stored at 15 ℃ for 0, 24, 60 and 96 h (labeled S0, S24, S60 and S96, respectively).
Point 5: line 115-117: Very vague description. Please describe in more detail. In fact, there is nothing about statistics here.
Response 5: We have rewritten the paragraph as follow: The original data generated by the mass spectrometer were checked and identified using Proteome Discoverer 2.2 (Thermo Fisher Scientific, MA, USA). The search database was the V. volvacea database (https://mycocosm.jgi.doe.gov/Volvo1/Volvo1.home.html). This database was used for protein retrieval. A global false discovery rate (FDR) based on a fit ≤ 1% and peptides ≥ 2 were the criteria for a reliable protein evaluation. Based on the selected reliable protein, student’s t-test was used to identify statistically significant changes in proteins (P < .05). Fold changes (FC) were calculated relatively in two comparison groups (S96, S60, S24 compared to S0 ). For each comparison, the protein with a FC > 1.5 (or < 2/3) was considered to be a differential abundance protein.
Point 6: line 126: water production? is this correct? If so, what do you mean exactly?
Response 6: Thank you for your kind reminder. I have rewritten this sentence ". Rot was becomed serious,, and browning was aggravated, among other changed."
Point 7: figure 5 is too small and impossible to read. It is almost useless like this.
Response 7: Thank you for your kind suggestion.The figure 5 has been modified.
Point 8: line 232-233: why is this an attractive result?
Response 8: I have rewritten this sentence and deleted "which was also an attractive result"
Point 9: line 280 (item 4.2): there is no discussion here!
Response 9: Yes, the previous results are summarized to prepare for further discussion.
Point 10: line 296-298: is this really surprising or unexpected?
Response 10: This is related DEPs for study. After analyzing and reading the literature, it is believed that there are several differential proteins closely related to this study.
Point 11: line 307: which protein is PAB1? function?
Response 11: I have rewritten this PAB1 and added "Binds the poly(A) tail of mRNA. Appears to be an important mediator of the multiple roles of the poly(A) tail in mRNA biogenesis, stability and translation. In the nucleus, interacts with the nuclear cleavage factor IA (CFIA), which is required for both mRNA cleavage and polyadenylation. Is also required for efficient mRNA export to the cytoplasm. Acts in concert with a poly(A)-specific nuclease (PAN) to affect poly(A) tail shortening, which may occur concomitantly with either nucleocytoplasmic mRNA transport or translational initiation. Regulates PAN activity via interaction with the stimulator PAN3 or the inhibitor PBP1. In the cytoplasm, affects both translation and mRNA decay. "
Point 12: line 310-311: again not really surprising or unexpected.
Response 12: I have rewritten this sentence and deleted "Eukaryotic cells undergo a series of stress reactions upon exposure to physical, chemical and nutritional stress, during which the posttranslational modification of mRNA is regulated."
Point 13: figure 9 is too small, low quality and impossible to read. It is almost useless like this.
Response 13: I have deleted figure 9.
Point 14: line 336: not sure of what closure means.
Response 14: I have deleted figure 9.
Point 15: line 350-352: what does this has to do with fruit senescence?
Response 15: I originally wanted to use this sentence to explain the relationship between protein expression, fatty acids, membrane and aging. I have rewritten this sentence .” The oleic acid content of kiwifruits tends to decrease after harvest, and the linoleic acid level increases significantly with increasing lipoxygenase activity and gene expression, which intensifies membrane lipid peroxidation and leads to an increase in cell membrane permeability, a loss of cell membrane function and the promotion of fruit maturation and aging.”
Reviewer 4 Report
The manuscript entitled “Comparative Proteomics Study on the Postharvest Senescence of Volvariella volvacea at 15 ℃” is well written with a sound scientific background. The findings of this manuscript concluded that protein synthesis and transcription, fatty acid synthesis and metabolism, and amino acid synthesis and metabolism are highly active during the postharvest senescence of V. volvacea. However, a few questions/changes are needed to improve the manuscript, which are given as follows. Thus, I recommended this manuscript for publication in the journal of fungi after the minor revision.
Ø Keep the title as “Comparative Proteomics Study on the Postharvest Senescence of Volvariella volvacea”. Kindly remove “at 15 ℃”
Ø Figure 1. (A, B and D) Changes in the hardness…. kindly check it and change it as follows,
Ø Figure 1. Changes in the hardness (A), fracturability (B), and phenotype of V. volvacea fruiting bodies during different storage periods (C). Each bar represents the mean ± SE of the individual samples, * represents P<0.05.
Ø The volcano plot is a versatile Figure illustrating FC and t-test results simultaneously. Rewrite this sentence for better clarity.
Ø Please explain what basis is the experiment sampling time at 24, 60 and 90 hours?
Ø Kindly rewrite the Figure 3 legend for better clarity.
Ø Change Figure 6 and 7 labels (S24-S0(A), S60-S0(B) and S96-S0(C) to S24-S0 (A), S60-S0 (B) and S96-S0 (C).
Ø Kindly present figure 10 at the end of the discussion section as a concluding remark, not in the conclusion section.
Ø The conclusion needs to improve with the research findings.
Author Response
Response to Reviewer 4 Comments
Dear Reviewers and Editor,
First, I want to thank you for reviewing our article. Your comments were very professional, and we appreciate all the noted weaknesses of our manuscript. We believe that the revised article will be more suitable after revision. Modifications have been made to the original text base on the comments provided by the reviewers. The reviewers’ comments and the responses are as follows:
Point 1: Figure 1. (A, B and D) Changes in the hardness…. kindly check it and change it as follows,
Response 1: We revised “Figure 1. (A, B and D) Changes in the hardness….” to “Figure 1. (A, B and C) Changes in the hardness….”
Point 2: Figure 1. Changes in the hardness (A), fracturability (B), and phenotype of V. volvacea fruiting bodies during different storage periods (C). Each bar represents the mean ± SE of the individual samples, * represents P<0.05.
Response 2: We revised “Figure 1. (A, B and C) Changes in the hardness, fracturability and phenotype of V. volvacea fruiting bodies during different storage periods. Each bar represents the mean ± SE of the individual samples, * represents P<0.05.” to “Figure 1. Changes in the hardness (A), fracturability (B), and phenotype of V. volvacea fruiting bodies during different storage periods (C). Each bar represents the mean ± SE of the individual samples, * represents P<0.05.”
Point 3: The volcano plot is a versatile Figure illustrating FC and t-test results simultaneously. Rewrite this sentence for better clarity.
Response 3:: We have rewritten the sentences as follow: The volcano plot is a versatile Figure. The horizontal axis represents the fold chang of the probe, and the vertical axis represents the significance of the probe (-log10 p-value).
Point 4: Please explain what basis is the experiment sampling time at 24, 60 and 90 hours?
Response 4: Based on our previous study[22], in which the changes of morphological and physiological indexes at 0, 12, 24, 36, 48, 60, 72 and 96 h were measured; four time points, namely, 0, 24, 60 and 96 h, were selected for this comparative proteomics study.
Point 5: Kindly rewrite the Figure 3 legend for better clarity.
Response 5: We revised “As shown in Figure 3, the DEPs increased with storage time. The 213 of 595 DEPs overlapped in the S24-S0, S60-S0 and S96-S0 groups (Figure 3A), among which 165 proteins were down-regulated (Figure 3B) and 48 proteins were up-regulated (Figure 3C). In the S24-S0 group, 54 up-regulated proteins (Figure 3C) and 170 down-regulated proteins (Figure 3B) were identified for a total of 224 DEPs (Figure 3A). In the S60-S0 group, 160 up-regulated proteins (Figure 3C) and 302 down-regulated proteins (Figure 3C) were detected for a total of 462 DEPs (Figure 3A). A total of 207 up-regulated proteins and 356 down-regulated proteins (Figure 3B) were identified in the S96-S0 group for a total of 563 DEPs (Figure 3A). ” to “As shown in Figure 3, the DEPs increased with storage time. In the S24-S0 group, 54 up-regulated proteins (Figure 3C) and 170 down-regulated proteins (Figure 3B) were identified for a total of 224 DEPs (Figure 3A). In the S60-S0 group, 160 up-regulated proteins (Figure 3C) and 302 down-regulated proteins (Figure 3C) were detected for a total of 462 DEPs (Figure 3A). A total of 207 up-regulated proteins and 356 down-regulated proteins (Figure 3B) were identified in the S96-S0 group for a total of 563 DEPs (Figure 3A). ”
Point 6: Change Figure 6 and 7 labels ( to .
Response 6: We revised “S24-S0(A), S60-S0(B) and S96-S0(C)” to “S24-S0 (A), S60-S0 (B) and S96-S0 (C)”
Point 6: Line 232, change "S96-S0 DEPs" to " DEPs in S96-S0".
Response 6: We revised “S96-S0 DEPs involved 111 different metabolic pathways, 13 of which were significantly altered. to “DEPs in S96-S0 involved 111 different metabolic pathways, 13 of which were significantly altered.”
Point 6: Kindly present figure 10 at the end of the discussion section as a concluding remark, not in the conclusion section.
Response 6: Thank you for your kind suggestion.The figure 10 was Line 311
Point 7: The conclusion needs to improve with the research findings.
Response 7: We have made changes to the manuscript as follows: In conclusion, the data obtained in this study may help describe the mechanism of postharvest senescence of V. volvacea stored at 15 ℃. This mechanism may be related to the regulation of DEPs involved in protein synthesis and transcription, fatty acid synthesis and metabolism, and amino acid synthesis and metabolism. In addition, most RNA transport related proteins were down regulated. which may lead to decrease in related gene regulation. Other important proteins, such as fatty acid metabolism protein increased, changes in fatty acid composition affect the cell membrane, which may accelerate the ripening and perception of fruits. The study will provide a new perspective for studying the molecular mechanism of V. volvacea postharvest storage at 15 ℃.
Reviewer 5 Report
This is a well written paper.
Comments, questions, concerns:
1. Please define was iTRAQ is? The acronym is not defined.
2. At the end of the introduction, around line 77, it was not clear if you were describing a previous study or the current study.
3. Please provide more details on "hardness" and how it is quantified.
4. Figure 1 needs to say A, B and C not A, B and D
5. Figure 1 - why such variation on the mushrooms in figure C
6. Figure 5. The print is two small to see
7. Figure 6. Define "rich factor"
Author Response
Response to Reviewer 5 Comments
Dear Reviewers and Editor,
First, I want to thank you for reviewing our article. Your comments were very professional, and we appreciate all the noted weaknesses of our manuscript. We believe that the revised article will be more suitable after revision. Modifications have been made to the original text base on the comments provided by the reviewers. The reviewers’ comments and the responses are as follows:
Point 1: Please define was iTRAQ is? The acronym is not defined.
Response 1: We have rewritten the sentences as follow: “Advances in proteomics technology have enabled researchers to quantitatively measure proteomic changes in complex samples. In particular, the isobaric tags for relative and absolute quantitation (iTRAQ)-based quantitative proteomics has become a powerful method of choice because it covers a variety of proteins due to limitations in molecular weight, isoelectric point or hydrophobicity.”
Point 2: At the end of the introduction, around line 77, it was not clear if you were describing a previous study or the current study.
Response 2: We have rewritten the sentences as follow: Based on our previous study[22], in which the changes of morphological and physiological indexes at 0, 12, 24, 36, 48, 60, 72 and 96 h were measured; four time points, namely, 0, 24, 60 and 96 h, were selected for this comparative proteomics study
Point 3: Please provide more details on "hardness" and how it is quantified.
Response 3:: We have rewritten the sentences as follow: Hardness is the maximum force that the sample material resists pressing into its surface, which reflects the force on the V. volvacea without deformation. It is the most direct indicator of taste. V. volvacea samples with high hardness and low cohesiveness can show fracturability.
Point 4: Figure 1 needs to say A, B and C not A, B and D
Response 4: We revised “A, B and D” to “A, B and C”.
Point 5: Figure 1 - why such variation on the mushrooms in figure C
Response 5: We have rewritten the sentences as follow: With increasing storage time, the surface of the fruiting bodies browned. Rot was accompanied by water production, and browning was aggravated, among other changes. This may be caused by the loss of water in the fruiting body, the destruction of cell membrane and the decline of self-regulation during postharvest storage of Volvariella volvacea.
Point 6: Figure 5. The print is two small to see
Response 6: Figures 5 has been changed.
Point 7: Figure 6. Define "rich factor"
Response 7: We have rewritten the sentences as follow: The rich factor represents the ratio between the DEPs and all annotated proteins enriched in the pathway. Rich factor refers to the ratio of the number of differentially expressed proteins located in the go entry to the total number of transcripts located in the go entry in all annotated proteins. The greater the rich factor, the higher the degree of enrichment. The bubble scale represents the number of DEPs, and the depth of the bubble color represents the adjusted P value.
Round 2
Reviewer 3 Report
I acknowledge that the authors replied satisfactorily to my previous comments.
I have only a couple of minor revisions:
- I suggest changing the last sentence of the abstract to: “Therefore, our research provides a reference for further studies on the aging mechanism of V. volvacea.”
- Line 144/145: should be: “Rot became serious and browning was aggravated, among other changes”